# Children's Perception of Climate Change in North-Eastern Portugal

Ricardo Ramos [1,*], Maria José Rodrigues [1,*] and Isilda Rodrigues [2,*]

1   Research Centre in Basic Education (CIEB), Instituto Politécnico de Bragança, 5300-253 Bragança, Portugal
2   Education and Psychology Department, University of Trás-os-Montes and Alto Douro, 5000-801 Vila Real, Portugal
*   Correspondence: ricardo.ramos@ipb.pt (R.R.); mrodrigues@ipb.pt (M.J.R.); isilda@utad.pt (I.R.)

**Abstract:** Despite the impact that climate change is having on our planet and considering its consequences for future generations, much of the academic literature focuses on adolescent and adult perceptions, giving little relevance to children's perceptions. Children's voices have the potential to influence public opinion, which may in turn determine the direction of a new policy on the climate crisis. In this context, it is urgent that we understand how children perceive this problem. This quantitative study was based on the application of 245 questionnaires to children aged between 9 and 13 years old from five schools in north-eastern Portugal, more specifically in the region of Trás-os-Montes. We can say that this study was a convenience study because we delivered the surveys in the schools closest to the working area of the researchers. We used a questionnaire with 26 questions, 24 of which had closed responses (like the Likert type), one open response, and one with multiple choices. In this work, we conducted a descriptive and inferential statistical analysis, and prepared a database, using the statistical software IBM SPSS, which allowed us to conduct some statistical tests, selected according to variables. For the descriptive analysis, several parameters were used for the distribution of variables, namely, frequency, percentage, mean, and standard deviation. We rejected the null hypothesis (H0) and assumed for the inferential analysis that the sample does not follow a normal distribution, considering the fulfillment of the necessary criteria for parametric tests and after performing the Kolmogorov–Smirnov normality test, whose null hypothesis (H0) is that data are normally distributed, and given that the $p$-value for the variables under study was $p < 0.05$. In this regard, non-parametric tests were used. The Mann–Whitney test was used to compare the degree of agreement with climate change statements as a function of the student's gender and year of schooling, which is a non-parametric test suitable for comparing the distribution functions of an ordinal variable measured in two independent samples. The results show that most of the children expressed concern about the study's potential problem, and (42%) said they are concerned about climate change. However, they show some doubts and a lack of knowledge about some of the themes, like (33.5%) cannot name only one consequence of climate change. We also found differences between the two study cycles, with children in the 6th grade having a higher average in their understanding of the phenomenon ($p = 0.049$), as well as the level of education of the parents being positively correlated with a more ecocentric posture, we can see this when we considering the variable parents. We also found that 46.6% of the students say that television is where they learn more about climate change. From the results obtained, we can open new paths for future research and contribute to the definition of policies and educational practices since the school has the responsibility to cooperate in the production of values, attitudes, and pro-environmental behaviors.

**Keywords:** children; climate change; climate literacy; education; sustainable development

## 1. Introduction

There is a growing concern about how children understand climate change issues [1]. By having access to these studies on this subject, we can inform better educational op-

portunities and identify the best ways to address these challenges [2]. Education should contribute to mitigating the problem and reducing the vulnerability of people and communities faced with the consequences of climate change [3]. Education also respects sustainable development goals, especially number thirteen, which refers to climate change measures in national policies, strategies, and planning, i.e., improve education and raise awareness and human and institutional capacity on climate change mitigation, adaptation, impact reduction, and early warning measures. The same document said that children should be involved in solving problems also close to them, which may be problems of the region [4]. Allegedly, there seems to be a deficit in the ability of schools to sensitize students about this issue [5], as reported in a project carried out by the School Education Gateway [6], which showed that teachers in countries such as Spain, Turkey, Romania, and Canada do not have adequate skills to educate students about the climate crisis. Other studies that focused on understanding children's and adolescents' perceptions of the causes of climate change found that these tended to be unclear [7–9]. In other authors' works, they found several gaps in knowledge and undervalued the role of individual actions, showing that they felt impotent in relation to the issue in question [10,11]. Some authors also point out that children in these age groups often associate the ozone layer with climate change, and have difficulty enumerating causes, consequences, and solutions [1,12,13]. Children mostly depend on the information given to them by adults in various contexts, including at school and in mass media. As a result, they develop misunderstandings and confusion because of the misinformation they may be receiving [11,14]. In some of the textbooks, the mechanisms of climate change are often poorly explained or unclear in the way they present the information. This leads to doubts that reportedly seem to persist throughout adolescence and into adulthood [15]. A lack of training among journalists, and a lack of time to investigate a story and its background can therefore act detrimentally to the translation of science into information [16].

Addressing climate crisis-related content directly in preschool education could be an effective way to help break the cycle of gaps in adulthood. We consider that climate change education is an area that aims at designing and developing educational responses based on informed decisions that are effective in the context of the climate crisis. The Belgrade Charter, a reference in environmental education, also states that environmental action should improve all ecological relations, including the relations of human beings among themselves and with the other elements of nature, as well as develop a world population aware of and concerned about the environment and the problems associated with it, with the knowledge, ability, motivation, attitude, and commitment to act individually and collectively in the search for solutions to current problems and the prevention of new problems [17]. It follows that such decisions should be consistent with the goals of mitigation and adaptation to the inevitable consequences of a changing climate. We argue that climate change education should include a sense of urgency, emphasizing the long-term, systemic, complex, and unpredictable effects on the biosphere. In this sense, the education system should strengthen climate literacy, which means an understanding of your influence on climate and climate's influence on you and society, to enable society to mitigate this issue [18–20]. In this sense, we emphasize "education for sustainability," which is a concept that integrates the education process for the three pillars that constitute sustainability: environmental, social, and economic, creating a new way of thinking and acting that serves their needs without compromising the rights of future generations [4].

With this study, we want to know what level we are in terms of children's perceptions of this climate crisis. It is undeniable that schools must prepare students for societal challenges in the context of an uncertain or unpredictable future by encouraging the development of skills that enable them to question established knowledge, integrate emerging knowledge, communicate effectively, and solve complex problems.

Although climate education has gained significant attention in Europe and the rest of the world, Italy remains the only European Union country that has made climate change education compulsory in schools. The curriculum documents, called learning needs, for

these cycles of studies and students of these ages, contemplate that climate change is addressed [18]. But looking at the curriculum closely, in 2020, nearly half (47%) of the national curriculum frameworks of the 100 countries studied, including Portugal, did not mention climate change, and these curriculums included the first and second cycles. Of the few that addressed the topic, it was in a superficial and not very relevant way [21].

In this sense, teachers play a crucial role in empowering young people to develop their understanding of and attitudes toward climate education [5]. Like we said in the case of Portugal, the subject of climate change is part of the national curriculum (throughout compulsory education, i.e., from pre-school education to the end of secondary education [22], with this subject being addressed in the curricular areas of world knowledge, environmental studies, and citizenship, as shown in Table 1.

**Table 1.** Areas where climate change is discussed at school [18,19].

| Education Level | Study Area |
|---|---|
| 1º Cycle (9 years old)<br>2º Cycle (12 years old) | - Citizenship and Development<br>- Study of the Environment<br>- Natural Sciences |

In our study sample, in basic education, environmental issues are transversal, and according to the document Essential Learning (EA), they are addressed in the curriculum and can be articulated between other areas (Table 1).

We believe it is critical to understand primary and secondary school students' perceptions of climate change in the north-east of Portugal, to determine whether they take a more anthropocentric stance, characterize nature as something instrumental, or prioritize human control and dominion over nature with the goal of exploiting its resources. On the other hand, they may take a more ecocentric stance, in which they defend nature, even if, to do so, they give up their materialistic and consumerist lives [23]. We also set out to understand if their perceptions of climate change varied according to the following variables: age, gender, year of schooling, and parents' qualifications. This study also meets Sustainable Development Goals, objective number 13, which refers to the mitigation of climate change issues, and objective number 4, which refers to quality education. This theme is also addressed in the references to education for sustainable development and the national strategy for citizenship education [3].

## 2. Materials and Methods

The rationale of the current research paper is to explore the perceptions about climate change, how they are perceived by the 4th-grade students (first cycle), between 9 and 10 years old, and the 6th-grade students (second cycle), between 11 and 12 years old. As we mentioned before, the northeastern region is a region likely to face some of the consequences of climate change. The northeast of Portugal, where our study was focused, is a vulnerable zone, according to the Climate Risk Index (IRC). This is a region very connected to the agricultural sector, a sector of great relevance and essential for the structuring of the territory. As it is known, weather drastically affects agricultural productivity, making this a sector of great importance in the region [24,25]. As a result, we cover schools in the cold Transmontano land as well as those in the warm land. In the next Figure 1, the terra fria and terra quente regions are represented.

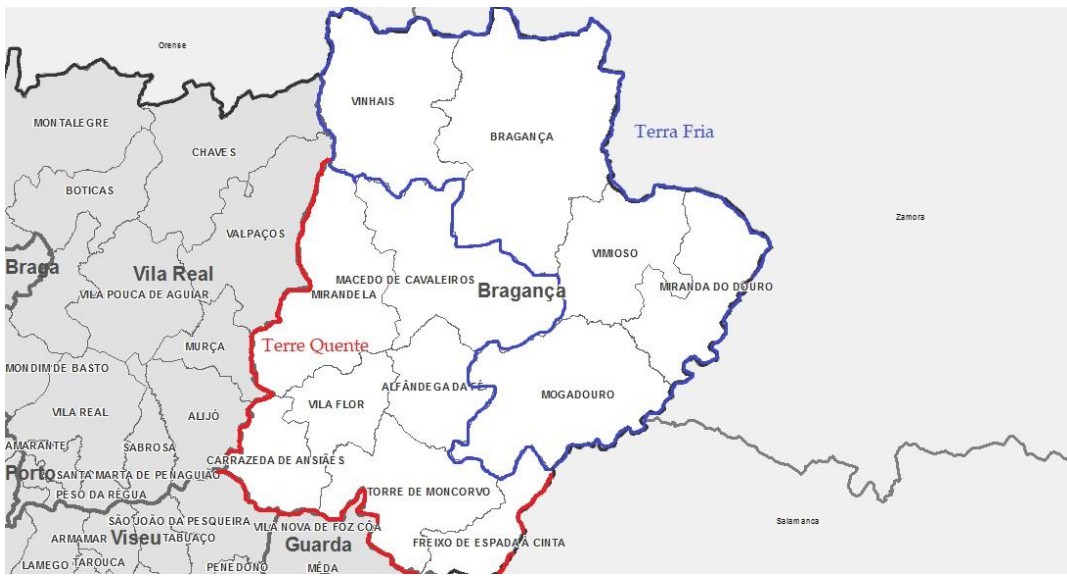

**Figure 1.** Map of the district of Bragança, with the regions of Terra Fria and Terra Quente. Source [24].

Of the five collaborating schools, two are inserted in the microclimate of warm land in the Transmontane (Vila Flor and Carrazeda de Ansiães) with a milder climate, marked by the Douro River valley and the valleys of its tributaries. The others, Vinhais, Macedo de Cavaleiros, and Bragança, are further north and are part of the regions of higher altitude, which constitute the Terra Fria Transmontana, where the landscape is dominated by higher mountains. In Table 2, we present the synthesis of our objectives and the questions we intend to answer.

**Table 2.** Research questions and research objectives.

| | Research Questions | | Objectives |
|---|---|---|---|
| 1. | What perceptions of climate change do first, and second-cycle students have? | 1. | Identify perceptions of the first and second-cycle students' perceptions of climate change. |
| 2. | Are these perceptions the same according to the variables—age, gender, year of schooling, and parents' qualifications? | 2. | To find out whether perceptions of climate change differ according to age, gender, year of schooling, and parents' education. |

The data collection instrument used was a questionnaire, which was previously approved by the Ethics Committee of the University of Trás-os-Montes and Alto Douro and the Directorate-General of Education of Portugal, (n°0811200001). It also had the consent of the student's parents and included an informed consent form stating that the students agreed to participate in the research, which was for academic purposes only. The questions in the instrument were written in an objective and simple way so that they would be easy for the children to understand. The questionnaire included 24 closed-ended items using a Likert scale with five response options (totally disagree, disagree, indifferent/don't know, agree, totally agree), it also contained one open-response question and a final multiple-choice question. Concerning this, we intentionally used five questions as shown in this paper. We focused on questions that could assess: (i) scientific understanding of the climate change phenomenon; (ii) consequences of climate change; and (iii) anthropocentric vs. ecocentric posture, as we can see later.

The questionnaire was applied during the months of March to May 2022 to 4th grade students, this being the last year referring to the 1st cycle and the 6th grade referring to the last year of the 2nd cycle, in five schools in northern Portugal, in the region of Trás-os-Montes. The survey occurred in the classroom with the investigator, teachers, and students.

As we said this study was a convenience study because we delivered the surveys in the schools closest to the working area of the researchers. Next, we will present the results that will allow us to characterize, briefly and empirically, in each school, the 4th and 6th-grade students who volunteered, describing a total of 242 students, as shown in Figure 2 below:

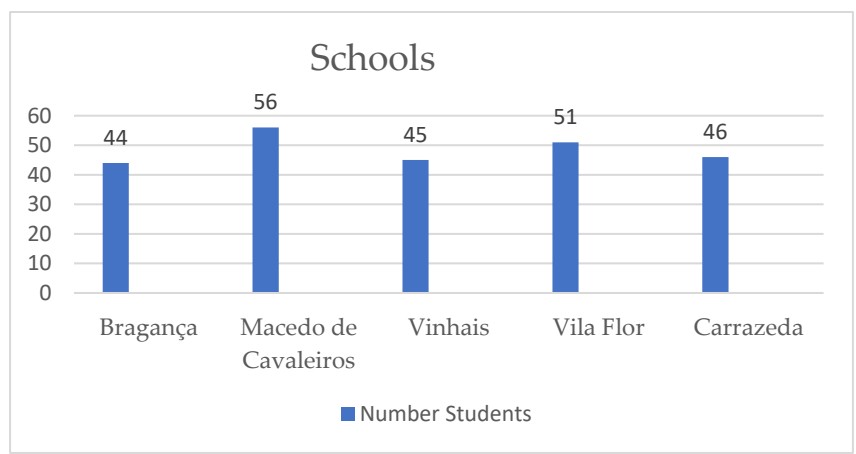

**Figure 2.** The number of students per school. N = 242.

The school in which more students participated in the study was Macedo de Cavaleiros, with 56 pupils responding (n = 56), followed by Vila Flor (n = 51), Carrazeda de Ansiães (n = 46), Vinhais (n = 45) and finally Bragança (n = 44). The different number of response is mainly since the size of the classes varies in the different school groups. Regarding the sample, the participating students are aged between 9 and 13 years, with an average age of 10 years, 113 in the 4th year, and 129 in the 6th year. As far the gender distribution is concerned, 43% are female and 56% are male. The average time the students took to answer the questionnaire was between 14 and 24 min. It is important to note that the researcher always used the questionnaire in a classroom setting with the teachers present, to make sure that the kids did not feel uncomfortable with the scenario and, simultaneously, could focus while completing it.

*Data Analysis*

Before applying the instrument, we conducted a pilot study with 12 students from another school, not included in this study, and two social sciences professionals, to test its reliability in academic terms. Regarding data treatment, we carried out a descriptive and inferential statistical analysis. We prepared a database using the statistical software IBM SPSS v27.0 (SPSS, IBM Corporation, New York, NY, USA), which allowed us to conduct some statistical tests, selected according to variables. For the descriptive analysis, several parameters were used for distribution of variables, namely frequency, percentage, mean, and standard deviation. We rejected the null hypothesis (H0) and assumed for the inferential analysis that the sample does not follow a normal distribution, considering the fulfillment of the necessary criteria for parametric tests and after performing the Kolmogorov–Smirnov normality test, whose null hypothesis (H0) is that data are normally distributed, and given that the $p$-value for the variables under study was $p < 0.05$. In this regard, non-parametric tests were used. The Mann–Whitney test was used to compare the degree of agreement with climate change statements as a function of the student's gender and year of schooling, which is a non-parametric test suitable for comparing the distribution functions of an ordinal variable measured in two independent samples. To correlate the degree of agreement with the climate change statements, the student's age, *nd* the parent's education, Spearman's correlation coefficient were used, which is a non-parametric measure of association between two at least ordinal variables. This coefficient is obtained by replacing the values of the observations with their respective

orders. Association measures quantify the intensity and direction of the association between two variables.

## 3. Results

### 3.1. Level of Concern in the Sample, about Climate Change

Table 3 shows the results of the question: "Climate change does not worry me." 41.7% totally agree, 30.6% disagree, 10.7% say it is not relevant, 12.4% agree that climate change does not worry them, and 4.5% totally agree that climate change is something that does not worry them.

**Table 3.** Results on concern about climate change.

| | N = 242 | |
| --- | --- | --- |
| | **Frequencies** | **Percentage** |
| Totally Disagree | 101 | 41.7% |
| Disagree | 74 | 30.6% |
| Not Relevant | 26 | 10.7% |
| Agree | 30 | 12.4% |
| Totally Agree | 11 | 4.5% |

Even though most of the respondents show some concern about climate change, we think that the number of those who are not concerned is considerable. Climate change is a climate crisis, an urgency, and all children should be sensitive to this problem.

Regarding the understanding of the phenomenon, when it comes to mentioning a consequence of climate change, the frequency and percentage are presented in the following table (Table 4) more at front as we can see later.

**Table 4.** Survey results of the consequences of climate change. N = 242.

| Consequences | Frequencies | Percentage |
| --- | --- | --- |
| I don't know | 81 | 33.5% |
| Extinction | 30 | 12.4% |
| Pollution | 25 | 10.3% |
| Sea Level | 17 | 7% |
| Don't have Water | 15 | 6.2% |
| Drought | 13 | 5.4% |
| High Temperature | 8 | 3.3% |
| Volcano Activity | 7 | 2.9% |
| More Fires | 6 | 2.5% |

### 3.2. List the Consequences of Climate Change

For the question asking students to list at least three consequences of climate change, we present the results in the word cloud, represented in Figure 2 below.

As a result of Figure 3, aided by the previous table number 5, we intend to know the student's understanding of the phenomenon, namely its consequences. We can see that 33.5% (n = 81) of the students in the sample cannot enumerate a consequence of climate change, answering that they did not know, even though school programs for these ages include the topic of climate change. To clarify this aspect, the extinction of flora and fauna, with a representation of around 12% (n = 30), is the second most mentioned consequence by the responding students, followed by pollution, represented by 10.3% (n = 25).

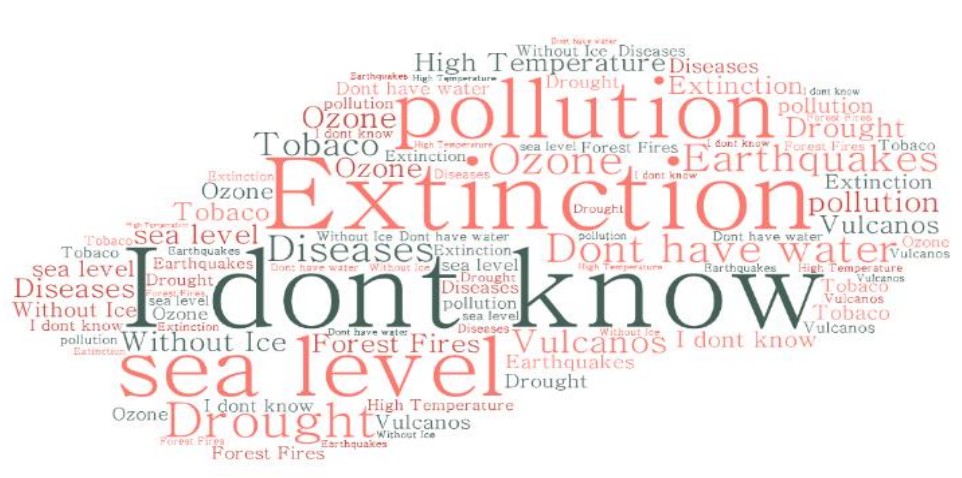

**Figure 3.** Word cloud on students' responses to the question "List one consequence of climate change" n = 242.

Other causes, although in a lower percentage, were mentioned by the students, such as rising sea levels, water shortages, and droughts, among others. Furthermore, we took note of the ambiguity of words for students, such as "consequence," and the investigator inquired about it during data collection, and all the students understood the word.

### 3.3. Male vs. Female Perception of Climate Change

Next, we discuss the degree of agreement with the statements on climate change, according to gender, present the significance value (Table 5), and then present the parameters of the results that proved to be statistically significant (Table 6).

**Table 5.** Significance value of the comparison of the degree of agreement with statements on climate change according to gender.

|  | **Male** | | | **Female** | | | *dif* | *p* |
|---|---|---|---|---|---|---|---|---|
|  | **N** | **Average** | **Dp** | **N** | **Average** | **Dp** |  |  |
| **I would like to learn more about climate change** | 105 | **3.59** | 1.14 | 137 | **3.93** | 1.04 | 0.34 | 0.014 |

**Table 6.** Significant results of the comparison of the degree of agreement with the climate change statements according to gender (*p* = Test Mann–Whitney).

|  | *p* |
|---|---|
| I would like to learn more about climate change | **0.014** |
| *Variable: **Gender "Female"** | (p = Test Mann–Whitney)* | |

The average level of agreement was higher among females. It is worth remembering that, regarding the distribution of the sample by gender, 43% are female and 56% are male, and even though the number of female students was smaller, a higher percentage of female students replied that they want to learn more about the problem of climate change.

*3.4. Perceptions of Climate Change 4th Year of Schooling vs. 6th Year of Schooling*

We prepared a cross-tabulation (Table 7) to see if there were differences in relation to the year of schooling and the question: Climate change is caused by humans, but it is also a natural phenomenon.

**Table 7.** Frequency table, grade 4 vs. grade 6, regarding the question: Climate change is caused by humans, but it is also a natural phenomenon. N = 242.

| | 4º Grade | | | 6º Grade | | | | |
|---|---|---|---|---|---|---|---|---|
| | **N** | **Average** | **Dp** | **N** | **Average** | **Dp** | *dif* | *p* |
| Climate Change is caused by humans, but it is also a natural phenomenon | 113 | **3.37** | 1.1 | 129 | **3.65** | 1.12 | 0.28 | 0.049 |

In this question, where we intended a scientific understanding of the phenomenon, it was possible to verify that the average was higher in the 6th grade in the statement "Climate change is caused by humans, but it is also a natural phenomenon" ($p = 0.049$).

*3.5. Perceptions on Climate Change according to the Variable "Education of the Parents"*

Next, we tried to find out if the parents' education could influence the students' perceptions of the causes of climate change. The family is not the most important factor in improving climate literacy, compared to school and mass media, for example [26]. But we wanted to know if there was any correlation between parents' studies and students' perceptions.

The next Table 8 gives an overview of the student's answers, considering the variable parents' education and the question "Our way of life, e.g., traveling, buying, and consuming overloads ecosystems" (r = 0.207).

**Table 8.** The student's degree of agreement and the school age of the parent with more education was correlated by Spearman's coefficient.

| | Age (Student) | Parents Education Level |
|---|---|---|
| Our way of life, e.g., traveling, buying, and consuming overloads ecosystems | 0.110 | 0.207 |

As we can see from Table 8, the educational level of the guardian is positively correlated with the degree of agreement with the statement "Our way of life, such as traveling, buying, and consuming overloads ecosystems." Although the intensity of the variables is a weak correlation (r = 0.207 *), which is significant.

*3.6. Distribution of Averages in Relation to the Question "Where Do You Learn More about Climate Change?*

In a final multiple-choice question, we wanted to find out where students considered learning more about climate change. The answers are represented in the next Figure 4, varied, indicating school, TV, the Internet, radio, family, books, or others. The answers are represented in Figure 4.

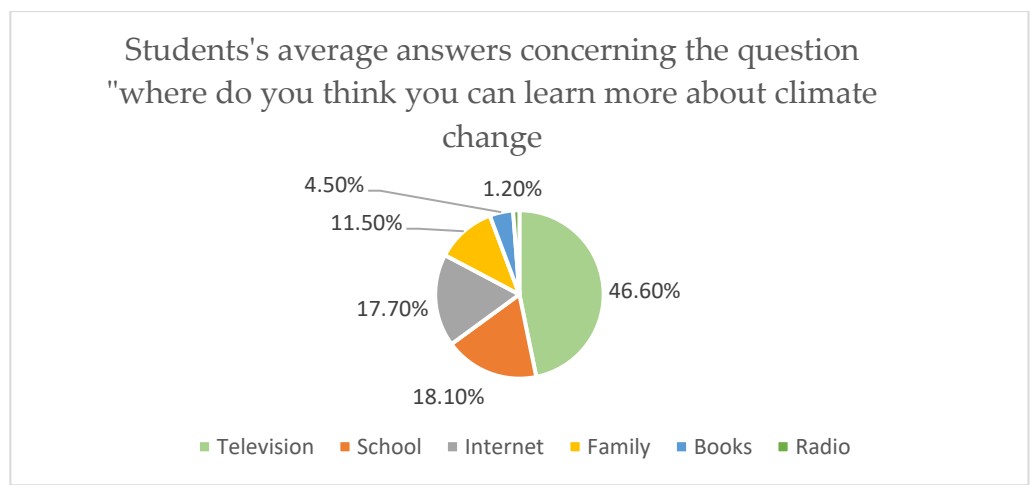

**Figure 4.** Distribution of average respondents' answers to the question "Where do you think you can learn more about climate change?".

By analyzing the graph, we can see that 46% of the students answered that it was on television, followed by 18% at school, 17% on the Internet, 11.5% in the family, 4.5% in books and finally 1.24% on the radio.

## 4. Discussion

We think that we have proven what other studies show that there is indeed a gap regarding climate literacy in European and American schools [6–8]. It was possible to verify that 70% of the students were concerned about climate change. However, their knowledge of the phenomenon falls short. Regarding the understanding of the phenomenon, when analyzing Table 3, we found that 33.5% could not list a consequence of climate change. Others mentioned pollution (10.3%) and volcanic activity (2.9%) because of climate change, when these should be mentioned as a cause, not a consequence. However, according to the literature, this type of response appears in other studies, in which respondents confuse natural geological phenomena with the consequences of climate change due to anthropogenic causes, demonstrating low levels of climate literacy [18,27]. At a time when more should be known about climate change, there face many confusions, misconceptions, naive theories, beliefs, and distorted perceptions, evidencing low levels of climate literacy. This fact concerns children, young people, and adults and constitutes ideas that often become widespread due, in part, to the abundance of complex, divergent, inaccurate information or even errors that are propagated in an era of mass information [28]. As far as verifying whether the perception of climate change could vary according to the parents' academic qualifications is concerned, we found that the level of education of the parents is positively correlated with the degree of agreement with the statement regarding the perception of one of the causes of climate change. As for the gender variable, we found that female students showed greater sensitivity than male students. In fact, in another study, [29] it was found that women, more often than men, are more sensitive to climate change and are more willing to change their behavior to reduce carbon dioxide emissions. However, there is no consensus in the available literature, as there are countries where women are more concerned than others, and there are others where men seem to be more concerned about environmental issues. Some of these differences may be attributed, for example, to differences in access to education [29]. It will therefore be a priority for all countries to provide climate change responsive education, as well as access to information on climate impacts for all population groups [30]. We discovered that the sixth grade has a higher average understanding of the phenomenon than the fourth grade. This can be attributed to the fact that older students have an easier time grasping certain concepts on the subject, as well as the fact that their schooling, the incidence of the topic in their daily lives in the classroom, ends up enriching their understanding of the topic [31]. Another



issue is related to the fact that most of the students stated that school is not the place to learn more about climate change. However, students pointed out that they also learned about climate change from TV (46.6%) and the Internet (17.7%). While we should praise the merit of some media in providing information, we should remember that such information is not always reliable or of good quality. They often use overly dramatic and catastrophic scenarios, sometimes even making "guesses" that are unlikely to happen and have little scientific basis, which can mislead the viewer [32,33]. Scientific information about the global climate and its development is complex. Mass media, such as the Internet, are the main sources of information on climate change, although in some situations they can negatively affect levels of knowledge, problem awareness, and behavioral intentions [34,35]. Children and young people are growing up in a globalized digital world in which they process information from a wide variety of sources. They must have critical literacy skills to navigate the potential "pitfalls" when consuming news, especially when using online sources and social media [32]. We agree that the education system should try to make climate education more effective. Climate change education can have two parts: climate and change. "Climate" they explain, involves the natural sciences, while "change", or educating for change, involves engaging the social sciences and humanities [36]. Persisting with traditional content delivery or a "business-as-usual" approach is increasingly seen as inadequate [37]. This being said, we argue that the education system should adapt and perhaps reinvent itself to enrich climate literacy and train more responsible and critical-minded citizens. This work, intended to collect data on children's perceptions, is only one part of a larger work, where in the future, we intend to implement activities aimed at enriching children's climate literacy. After that, we can interview teachers and students to get feedback on these activities and make the triangulate study.

## 5. Study Limitations

This study has several limitations, and the results should be interpreted with those limitations in mind. The main limitation is related to the sample, which covered only one region of northern Portugal. Furthermore, we believe that our survey is a blunt tool for this type of topic, such as climate change, but on the other hand, we want to know our children's background on this phenomenon, even if it is a short and simple vision, to create a good starting point to understand how our students are concerned with this topic.

## 6. Conclusions

This study proved that although climate change concerns students and they want to know more about it, some misconceptions still persist. Since this topic is highlighted as a priority and fundamental for the resolution of this emergency, there should be more efforts toward climate change education. We hope, therefore, that this work will contribute to rethinking educational systems, more pro-environmental so that students can exercise a more active, coherent, and responsible citizenship. We also believe that climate change should have more expression in curricula and educational resources, which may also include the initial and continuing training of educators and teachers. We suggest that climate literacy should go beyond the natural sciences and be integrated into the various subject areas, from the social sciences (economics, sociology, anthropology, geography, etc.,) to the humanities (philosophy, ethics, etc.,) [30]. As we can see, there is a correlation between the parents' academic qualifications and knowledge about climate change, so governments should also invest in education for all, meeting the objectives of sustainable development. Schools, we believe, should play an important role in raising awareness and educating about climate change. As an institution responsible for the comprehensive training of citizens, schools, from early childhood education to higher education, must embrace climate education for greener progress and view this issue as a serious problem, considering it the great challenge of the 21st century. For over a decade, we have continued to educate young people as if there is no planetary emergency [38]. Teachers face a challenge

in providing climate education. In the future, we will not have the luxury of choice. This and other studies of this nature are important in giving us signals that we must act quickly.

**Author Contributions:** Conceptualization, R.R. and M.J.R.; methodology, I.R.; software, R.R.; validation, I.R., M.J.R.; formal analysis, R.R.; investigation, R.R.; resources, R.R.; data curation, R.R.; writing—review and editing, M.J.R.; visualization, I.R.; supervision, I.R. All authors have read and agreed to the published version of the manuscript.

**Funding:** This work has been supported by FCT—Fundação para a Ciência e Tecnologia within the Project Scope: UIDB/05777/2020.

**Institutional Review Board Statement:** The study was conducted in accordance with the Declaration of Helsinki, and approved by Ethics Committee) Directorate-General of Education of Portugal (0811200001) for studies involving children.

**Informed Consent Statement:** Informed consent was obtained from all subjects involved in the study.

**Data Availability Statement:** Not applicable.

**Conflicts of Interest:** The authors declare no conflict of interest.

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
