# Peer review of "Children’s Perception of Climate Change in North-Eastern Portugal"

_societies, doi:10.3390/soc13010006_

Round 1

Reviewer 1 Report

Please see the attached document with my highlighted comments.

Author Response

Response to Reviewer 1 Comments

Dear reviewer;

Thank you for your suggestions and revisions, which we have read carefully. We are going to put all the changes we made in our document. Please consider that some changes in the original word, are about the other reviewer suggested to us.

Point 1: In the abstract, we listened to your words and we put more statistical data because it is a quantitative study; 

“We used a questionnaire with 26 questions, 24 of which had closed responses (like the Likert type), one open response, and one with multiple choices. In this work, we make conduct a descrip-tive and inferential statistical analysis, and prepared a database, using the statistical software IBM SPSS, which allowed us to conduct some statistical tests, selected according to variables. For the descriptive analysis, several parameters were used for the distribution of variables, namely, frequency, percentage, mean, and standard deviation. We rejected the null hypothesis (H0) and assumed for the inferential analysis that the sample does not follow a normal distribution, con-sidering the fulfillment of the necessary criteria for parametric tests and after performing the Kolmogorov-Smirnov normality test, whose null hypothesis (H0) is that data are normally dis-tributed, and given that the p-value for the variables under study was p< 0.05. In this regard, non-parametric tests were used. The Mann-Whitney test was used to compare the degree of agreement with climate change statements as a function of the student's gender and year of schooling, which is a non-parametric test suitable for comparing the distribution functions of an ordinal variable measured in two independent samples. The results show that most of the chil-dren expressed concern about the study's potential problem, and (42%) said they are concerned about climate change. However, they show some doubts and a lack of knowledge about some of the themes, like (33,5%) cannot name only one consequence of climate change. We also found dif-ferences between the two study cycles, with children in the 6th grade having a higher average in their understanding of the phenomenon (p = 0,049), as well as the level of education of the parents being positively correlated with a more ecocentric posture, we can see this when we considering the variable parents. We also found that 46.6% of the students say that television is where they learn more about climate change. From the results obtained, we can open new paths for future re-search and contribute to the definition of policies and educational practices since the school has the responsibility to cooperate in the production of values, attitudes, and pro-environmental be-haviors.”

Point 2: In the introduction, you recommended the following: "Sensitisation alone cannot change attitudes and behaviours. See Education for Sustainability under UNESCO" we put an excerpt from unesco education, also mentioning that children should be involved in solving problems identified in the region. “Education also respects sustainable development goals, especially number thirteen, which refers to climate change measures in national policies, strategies, and planning. Improve education and raise awareness and human and institutional capacity on climate change mitigation, adaptation, impact reduction, and early warning measures. The same document said that children should be involved in solving problems also close to them, which may be problems of the region [4].”

Point 3: On line 70 of the introduction. We have included a chapter on Belgrade 75, as you mentioned.

Point 4: In line 96, you mention:This is vague. In what way/s?: we reinforce the idea, where we point out the Portuguese curricula, contemplate climate change, but when the sutdy of Unesco look closely, it falls short of expectations.

Point 5: In the materials and methods, we have included a map of the region where we intervened, as you suggested. 

Point 6: In Figure 2, I have put more details, as you suggested. I put that this is a convenience sample because they were schools close to where the researchers work. And we put more details.

Point 7: Table 8: we added the following: he family is not the most important factor in improving climate literacy, compared to school and mass media, for example. But we wanted to know if there was any correlation between parents' studies and students' perceptions.

Point 8: In the discussion you mention: This study is incomplete. You need to condcut a curriculum audit to triangulate your findings, as well as interview teachers. We add:This work, intended to collect data on children's perceptions, is only one part of a larger work, where in the future, we intend to implement activities aimed at enriching children's climate literacy. After that, we can interview teachers and students to get feedback on these activities and make the triangulate study.

I hope we have met your expectations. 

If there is something missing, please do not reject the article, we will be happy to review and read your suggestions.

Reviewer 2 Report

Dear authors,

Thank you for your manuscript. The study includes interesting results.

I have a few minor suggestions (cited below and highlighted and commented in the attach pdf file) after carefully reading your manuscript:

Authors should revise numbering of the "results". Errors are highlighted in the attached pdf document.

I found that the conclusions are somewhat detached form the actual results presented. Authors conclude that climate literacy is below the desired level but it is not specified how this is measured for this particular study. The relation between climate literacy and gender and academic qualifications is not mentioned in the conclusions. Rather, the conclusions are a reflection of what Climate Change Educations should look like but considering the results presented I find this out of context. The manuscript could be concluded with this reflection but the conclusions should also make reference to all the aspects of the data presented.

Author Response

Response to Reviewer 2 Comments

Dear reviewer;

Thank you for your suggestions and revisions, which we have read carefully. We are going to put all the changes we made in our document. Please consider that some changes in the original word, are about the other reviewer suggested to us.

Point 1: We have revised the numbering of the graphs;

Point 2: In the conclusion you say "The authors conclude that climate literacy is below the desired level, but they have no way to measure it." we found your recommendation interesting, so we modified the text so that this work will contribute to rethinking educational systems, more pro-environmental so that students can exercise a more active, coherent, and responsible citizenship.

Point 3: You also mention in the conclusion "The relationship between climate literacy and parental qualifications, is not mentioned in the conclusions" we added that part. (…) “As we can see, there is a correlation between the parents' academic qualifications and knowledge about climate change, so governments should also invest in education for all, meeting the objectives of sustainable development.”

I hope we have met your expectations. 

If there is something missing, please do not reject the article, we will be happy to review and read your suggestions.

Round 2

Reviewer 1 Report

Kindly refer to the comments that are typed on the document.

Author Response

Dear reviewer.

The document you have attached seems to be the suggestions of the first phase, to which we have already responded. We have attached the answers again. We also send the final article with the corrections you suggested 

Please let us know if we should do any more repairs to the article.

We would be happy to follow your suggestions.
Best regards
